# Vitamin E Lipid-Based Nanodevices as a Tool for Ovine Sperm Protection against Oxidative Stress: Impact on Sperm Motility

**DOI:** 10.3390/antiox11101988

**Published:** 2022-10-06

**Authors:** Alejandro Jurado-Campos, Pedro Javier Soria-Meneses, María Arenas-Moreira, Carlos Alonso-Moreno, Iván Bravo, Virginia Rodríguez-Robledo, Irene Sánchez-Ajofrín, Ana Josefa Soler, José Julián Garde, María del Rocío Fernández-Santos

**Affiliations:** 1SaBio IREC (CSIC—UCLM—JCCM), Campus Universitario, s/n, 02071 Albacete, Spain; 2Centro Regional de Investigaciones Biomédicas, Unidad NanoDrug, 02071 Albacete, Spain; 3Facultad de Farmacia, Universidad de Castilla la Mancha, 02071 Albacete, Spain

**Keywords:** nanoemulsions, vitamin e, nanotechnology, sperm oxidative stress, drug liberation

## Abstract

The advent of nanotechnology in the field of animal reproduction has led to the development of safer and more efficient therapies. The use of nanotechnology allows us to avoid the detrimental effects of certain traditional antioxidants, such as Vitamin E. Its hydrophobic nature makes mandatory the use of organic solvents, which are toxic to sperm cells. This study aims to evaluate the efficiency of vitamin E nanoemulsions (NE) on ram (*Ovis aries*) spermatozoa. For this purpose, the effect of three NE concentrations (6, 12, and 24 mM) were assessed on sperm of 10 mature rams of the Manchega breed. Sperm samples were collected by artificial vagina, pooled, and diluted in Bovine Gamete Medium. The samples were stored at 37 °C and assessed at 0, 4, 8, and 24 h under oxidative stress conditions (100 µM Fe^2+^/ascorbate). Motility (CASA), viability (YO-PRO/IP), acrosomal integrity (PNA-FITC/IP), mitochondrial membrane potential (Mitotracker Deep Red 633), lipoperoxidation (C_11_ BODIPY 581/591), intracellular reactive oxygen species (ROS) production and DNA status (SCSA^®®^) were assessed. A linear mixed-effects models were used to analyze the effects of time, NE, and oxidant (fixed factors) on sperm parameters, and a random effect on the male was also included in the model with Tukey’s post hoc test. Protection of ram spermatozoa with NE resulted in a more vigorous motility under oxidative stress conditions with respect Control and Free vitamin E, while preventing the deleterious effects of oxidative stress coming from the production of free radicals and lipid peroxidation. These results ascertain the high relevance of the use of delivery systems for sperm physiology preservation in the context of assisted reproduction techniques.

## 1. Introduction

Spermatozoa are highly specialized cells responsible for fertilization by delivering paternal genome to MII oocytes [1]. To do so, sperm motility is an essential requirement. Spermatozoa make use of adenosine triphosphate (ATP) by flagellar dynein-ATPase localized into the axoneme [2] to flagellar movement. Even though ATP production can be accomplished by two pathways: aerobic metabolism which involves citric acid cycle and oxidative phosphorylation [3], and/or anaerobic metabolism through the glycolytic route [4], oxidative phosphorylation performed by mitochondria is the main source of ATP production required for flagellar movement. On this subject, several studies reported mitochondrial alterations related to the loss of sperm motility [5].

Reactive oxygen species (ROS) are by-products of the ATP synthesis. Appropriate concentration of ROS is associated with sperm motility [6,7], an excess amount are related to a reduction of mitochondrial activity [6,7], DNA oxidation, and fragmentation [8,9], whereas a deficit hampers motility. The balance between the production of ROS and the scavenging of ROS must be maintained to prevent oxidative stress [10]. On the other hand, plasma membranes of spermatozoa contain a large amount of polyunsaturated fatty acids prone to oxidation, which cause lipid peroxidation (LPO) with a detrimental effect on sperm motility [11,12,13]. In view of the above, many strategies have been pursued to reduce the deleterious effects of oxidative stress in recent years. Handling sperm samples on laboratory requires the use of antioxidants to preserve sources of stress, such as cryopreservation or sperm separating X- and Y through flow cytometry. In this very context, the use of a plethora of antioxidants and their combinations have been deeply assessed to maintain the balance between ROS production and antioxidant activity [14,15,16,17,18,19,20,21,22,23,24].

Over the past years, nanotechnology has raised many expectations to lead to significant improvements in health care [25,26,27]. In the context of assisted reproduction, several successful nanodevices have been reported [22]. Vitamin E (VE) is a powerful lipid-soluble antioxidant and extensively used to prevent oxidate stress. Nevertheless, its application in assisted reproduction is restricted due to the toxicity of the organic solvents required for administration [14,15,17,18,28]. The use of Vitamin E delivery systems, such as nanoemulsions [22,29] or hydrogels [29], offers several advantages in comparison to conventional administration. The beneficial effects of these nanodevices comprise improvement to sperm motility parameters with respect to free VE, reduction in ROS production and lipid peroxidation under oxidative stress conditions.

Recently, we reported lipid-based VE nanodevices as a tool for preserving red deer post-thawed sperm samples against oxidative stress [29]. This report was a pioneering work in assisted reproduction technologies in which vitamin E nanodevices were tested in ram spermatozoa, as a proof of concept. Herein, we expand the use of VE delivery systems for ovine sperm protection against oxidative stress. Data collection based on sperm motility, viability, mitochondrial activity, ROS production, lipid peroxidation and DNA damage support this strategy. This study provides valuable evidence to ascertain this strategy for stress protection on ram sperm.

## 2. Materials and Methods

### 2.1. Reagents and Media

Flow cytometry equipment, software, and consumables were purchased from Beckman Coulter (Fullerton, CA, USA). The fluorescence probe PI (propidium iodide), vitamin E (CAS number 10191-41-0), Trolox (analogue vitamin E), FeSO_4_·7H_2_O and L-α-Phosphatidylcholine from egg yolk were obtained from the Sigma Chemical Co. (Madrid, Spain). DiD oil (DiIC_18_(5) oil) and Hoechst 33342 were purchased from Invitrogen^TM^. The rest of the fluorescent probes were purchased from Thermo Fisher Scientific (Barcelona, Spain), unless otherwise stated. The stock solutions of the fluorescence probes were as follow: PI: 1.5 mM in Milli-Q water; YO-PRO-1: 50 μM in DMSO; PNA-FITC: 0.2 μM in DMSO; Mitotracker Deep Red 633: 1 mM in DMSO; C_11_ BODIPY 581/591: 0.2 mM in DMSO; CMH2DFCDA: 0.5 mM in DMSO. All the fluorescent stocks were kept at −20 °C, in the dark until needed.

Trolox solution (vitamin E analogue) was prepared from a 10 mM stock solution of ethanol. The oxidant solution was prepared as 10 mM FeSO_4_ and 50 mM sodium ascorbate (Fe^2+^) in water. Bovine gamete medium (BGM) and the freezing extender were prepared in our laboratory as described previously [21].

For nanoemulsion formulation, acetone (ACS reagent), ethanol (ACS reagent), Epikuron 145 V (20 mg) and Poloxamer 127 were purchased from Sigma-Aldrich (Madrid, Spain). Characterization of the nanodevices was achieved by Z-sizer NanoZS from Malvern Panalytical (Malvern, UK).

### 2.2. Vitamin E Nanoemulsion Formulation

Controlled emulsifications were employed to obtained VE nanoemulsion following previous reported procedures [22]. An ethanol/acetone (1:19) organic phase comprised of vitamin E and Epikuron 145 V (20 mg) was poured onto 10 mL of a 0.25 *v/w* aqueous solution of Poloxamer 127, and the mixture was stirred for 10 min. Finally, organic solvents were removed by reduce pressure to give rise to VE loaded nanoemulsions (NE) [30]. Size, polydispersity index, and Z-potential of nanoparticles were analyzed by using dispersion light scattering technique on a Zetasizer Nano ZS (Malvern Instruments, Malvern, UK) instrument. Data were analyzed using the multimodal number distribution software included with the instrument. Encapsulation efficiency of liposomes was calculated by using the difference of drug feeding and the non-encapsulated drug found in the supernatant after dialysis in phosphate buffered saline medium.

### 2.3. Animals

Animal experimentation was implemented following Ethics Committee University of Castilla-La Mancha. According to Spanish Animal Protection Regulation, RD 53/2013, which conforms to European Union Regulation 2010/63, we make the handling of the animals. A total of ten Manchega rams (>3 years of age), housed at the Experimental Farm of the University of Castilla-La Mancha (UCLM), were used in this study.

### 2.4. Semen Collection and Initial Evaluation

An ejaculate for each male, collected by using an artificial vagina, was used in this study. After sperm collection, the samples were carried at 37 °C to laboratory. Instantaneously, using a bright field and phase contrast microscopy (Eclipse 50i Nikon; Tokyo, Japan), volume (mL), mass and individual sperm motility (scale 1–5), percentage of motile sperm (%), and sperm concentration (spz/mL) were evaluated [31]. Sperm concentration was determined with a Makler counting chamber and the Sperm Class Analyzer software (SCA^®®^) (Microptic, Barcelona, Spain). The samples with a minimum of 70% motile sperm and wave motion of 3.5 were used for this study, measured subjectively.

### 2.5. Experimental Design

The aim was to examine the effect of three nanoemulsions (NE) concentrations (6, 12, and 24 mM) on sperm antioxidant protection of 10 mature rams of the Manchega breed. Moreover, two controls were employed. One of them was not supplemented (control) and the other was supplemented with Trolox 1 mM (free vitamin E), as a second control of the effect of dissolved vitamin E. Stock solutions of NE were diluting 1:1 with BGM 2× to a final concentration of 6 mM (NE6), 12 mM (NE12), and 24 mM (NE24) mM. The treatments were replicated, and the oxidant solution supplementation occurred at a final concentration of 100 μM Fe^2+^ and 500 μM ascorbate. Fe^2+^ oxidizes to Fe^3+^, which is recycled by the ascorbate, producing the highly reactive hydroxyl radical (HO). The sample was resuspended in each treatment up to 60 × 10^6^ m/mL.

The treatments were incubated at 37 °C and analyzed for sperm motility, apoptosis like-changes, lipid peroxidation, ROS production, and DNA damage after 0, 4, 8, and 24 h.

### 2.6. Sperm Motility Analysis

Sperm motility was evaluated using a Sperm Class Analyzer^®®^ (SCA V6.2, Microptic S.L; Barcelona, Spain) at 37 °C. Spermatozoa were diluted in TCF and loaded into a Makler counting chamber (10 µm depth). A minimum of 300 spermatozoa were analyzed per sample. For each sperm, the software rendered the following parameters: percentage of motile sperm (TM), percentage of progressive motile sperm (PM), velocity according to the actual path (VCL; µm/s), velocity according to the straight path (VSL; µm/s), velocity according to the smoothed path (VAP; µm/s).

### 2.7. Flow Cytometry Analysis

In our study, two flow cytometers were employed. Sperm viability and apoptosis-like changes, acrosome integrity, mitochondrial activity, lipid peroxidation and ROS production were carried out using a Cytoflex LX (Beckman Coulter, Inc., Brea, CA, USA) operated with the CytExpert version 2.3.0.84 (Beckman Coulter, Inc.). An FC-500 (Beckman Coulter, Brea, CA, USA) operated with the MXP software (v.3) was used to measure SCSA^®®^. IDEAS^®®^ software and WEASEL software (WEHI, Melbourne, Australia) were used to analyze raw data, respectively. All the fluorochromes were excited with a 488 nm laser, with the exception of Mitotracker Deep Red and Hoechst 33342, which were excited with a 633 nm helium–neon laser and with a 405 nm violet laser, respectively. A total of 10,000 events were acquired per sample. Dot plots with forward-scatter light and side-scatter light or aspect ratio and area were employed in the respective cytometers to exclude debris from the sperm population.

### 2.8. Fluorescence Probes

#### 2.8.1. Sperm Viability and Apoptosis-like Changes

Sperm viability and apoptotic-like status were evaluated with YO-PRO-1 and PI. Semen samples were analyzed by means of flow cytometry [32] after 15 min of dark incubation. Specifically, we evaluated the fluorescent DNA markers for cells with compromised plasma membrane. Three subpopulations were obtained: viable (unstained: YO-PRO-1-/PI−), apoptotic-like membrane changes (YO-PRO-1 +/PI−), and non-viable (membrane damaged: PI+) [10].

#### 2.8.2. Assessment of Acrosome Integrity

Using IP and PNA-FITC; acrosome integrity was assessed. This fluorescent technique allows classifying among four sperm populations: alive with intact acrosomes (PNA−/PI−), dead with intact acrosomes (PNA−/PI+), alive with damaged acrosomes (PNA+/PI−) and dead with damaged acrosomes (PNA+/PI+) [33] due to PNA (peanut agluttinin) binding specifically to the internal side of the external membrane of the acrosome, labelling them acrosome-damaged spermatozoa

#### 2.8.3. Assessment of Mitochondrial Activity

The mitochondrial status, investigated through the assessment of mitochondrial membrane potential (MMP), was evaluated using a Mitotracker Deep Red 633 fluorescent probe. The percentage of sperm Mitotracker+/YO-PRO-1-represented the proportion of viable sperm with active mitochondria [12]. The stained samples were incubated for 30 min in the dark before being run through the flow cytometer.

#### 2.8.4. Assessment of Lipid Peroxidation

Lipid peroxidation was estimated using the C11-BODIPY 581/591 fluorescent probe. An aliquot of each sample was incubated for 30 min with C11-BODIPY 581/591. The sperm samples were washed by means of centrifugation (600× *g*, 5 min) and were extended in BGM [17]. These steps were repeated at 4, 8, and 24 h and analyzed by using flow cytometry.

#### 2.8.5. Production of Reactive Oxygen Species

Reactive oxygen species production was recorded using the fluorescent probe 5-(and-6)-chloromethyl-2′,7′-dichlorodihydrofluorescein acetyl ester (CM-H_2_DCFDA). CM-H_2_DCFDA penetrates the plasma membrane and is retained after intracellular esterases cleave the acetate groups and emit green fluorescent (504 nm) upon oxidation [17]. The intensity of fluorescence of CM-H_2_DCFDA increases simultaneously as ROS production. The samples were analyzed, and the median H_2_DCFDA fluorescence of the viable sperm population value was noted [10].

#### 2.8.6. Sperm Chromatin Structure Assay

Following the SCSA*^®®^* (Sperm Chromatin Structure Assay), based on the susceptibility of sperm DNA to acid-induced denaturation in situ and on the subsequent staining with the metachromatic fluorescent dye acridine orange, chromatin stability was assessed. Acridine orange (AO) fluorescence shifts from green (dsDNA; double strand) to red (ssDNA; single strand) subject to the degree of DNA denaturation [34].

%DFI (% of spermatozoa with DFI > 25) and HDS (High DNA Stainability: % of spermatozoa with green fluorescence higher than channel 600, of 1024 channels) were obtained by analyzing flow cytometry data [35].

### 2.9. Confocal Laser Scanning Microscope Analysis

Confocal microscopy was used to visualize Vitamin E nanoemulsion uptake into sperm cells. DiD-labelled nanoemulsions were obtained following the protocol described by Sanchez-Rubio et al., 2020 [22] with modifications. Briefly, the nanoemulsions were obtained by employing controlled emulsification, when an organic phase comprised ethanol/acetone (1:19), vitamin E, Epikuron 145 V (20 mg), and DiD oil (80µL reconstituted onto ethanol) were poured onto 10 mL of a 0.25 *v/w* aqueous solution of Poloxamer 127. The solution was subjected to magnetic stirring for 10 min and the solvents were evaporated using a rotary evaporator. Finally, nanoemulsions were dialyzed in cassettes to remove the excess of DiD (tracer that has not been encapsulated).

To analyze the samples under confocal microscopy, spermatozoa were incubated with DiD-labelled nanoemulsions (30 min, RT, darkness). Then, samples were mounted on precleaned slides with a 1 µL drop of Slowfade™ from Thermo Fisher Scientific (Barcelona, Spain) and 5 µg/mL of Hoechst 33342. Fluorescent images were recorded using a Nikon A1 confocal microscope (Nikon Instruments Europe, Amsterdam, The Netherlands) at a 25× objective and under two channels to detect the nanoemulsions and the chromosomes. For that, lasers of 408 and 640 nm were used to excite Hoechst 33342 and DiD dyes, sequentially.

### 2.10. Statistical Analysis

Vitamin E treatment (Free VE, NE6, NE12, or NE24); oxidant; motility; viability; acrosome integrity; mitochondrial membrane potential; ROS and lipid peroxidation production; and DNA damage were analyzed by factorial ANOVA with IBM SPSS Statistics 25 (IBM, Armonk, NY, USA). Previous to statistical analysis, the assumption of normality and the equality of variances were checked out using the Kolmogorov–Smirnov normality test and a Levene’s test, respectively. A random effect on the male was also included in the model. When a significant effect was observed, post hoc Tukey’s corrections were carried out to adjust the *p*-values and account for multiple testing. Results are presented as mean ± SEM, and statistical significance was accepted for *p* < 0.05.

## 3. Results

### 3.1. Vitamin E Nanoemulsions: Formulation, Characterization and Confocal Microscopy

NE were obtained following published procedures [29] and characterized by using Dynamic light scattering (DLS) measurements. NE formulations displayed an average particle size of 159 ± 1.4 nm and a polydispersity index (PDI) of 0.118 ± 0.146. NE showed a ζ-potential of −18.3 ± 4.2 mV, which is indicative of a significant degree of stability. The high encapsulation efficiency of the formulation (98%) was determined by analyzing supernatant after 30 min dialysis in PBS. Figure 1 shows standard laboratory staining procedures for ram sperm. Nucleus of sperm head was stained by Hoechst 33342 (Figure 1A) and NE by DiIC18 (Figure 1B). From this image, spherical formulations were detected. It was not observed any significant changes to the size of the nanodevices after dispersion of the NE in BGM medium, according to previous stability studies reported [29]. Finally, Figure 1C shows spermatozoa marked with fluorescent combination, which verifies the uptake undertaken by NE.

### 3.2. Vitamin E Nanoemulsions Effects on Sperm Motility Assessed by CASA

Figure 2 and Figure 3 displayed the stimulatory effect of NE on kinematic sperm parameters (motility and velocity). Without exogenous oxidative stress, no significant differences were detected between Control and NE treatments. NE preserved total and progressive motility (Figure 2a) at values similar to the Control after 4 and 8 h, whereas free VE produced a significative decrease displaying a complete inhibition after 8 h of treatment (*p* < 0.001). Overall, the proportion of active spermatozoa decreased after 24 h-incubation (with and without exogenous oxidative stress). Therefore, sperm showed a similar percentage of total and progressive motility in all the treatments with not significant differences.

Under exogenous oxidative stress conditions, sperm showed a different behavior (Figure 2). NE improved total and progressive motility after 4 h of treatments. At 8 h, NE slightly improved the results of Control. However, a small percentage of motile sperm in the treatments with NE treatments was observed. As expected, sperm motility inhibition was achieved after free VE treatments at 4 h.

Speed parameters VCL, VSL, and VAP (Figure 3) did not significantly differ after incubation with NE treatments when comparing to Control but all of them showed substantial differences regarding to free VE at any time (*p* < 0.001). Nevertheless, under exogenous oxidative stress conditions the NE treatments improved the results revealed by Control in all the scenarios analyzed, following the pattern (NE 12 mM > NE 6 mM > NE 24 mM). Overall, free VE, with or without exogenous source of oxidative stress, decreased the speed parameters (*p* < 0.001).

### 3.3. Vitamin E Nanoemulsions Effects on Sperm Viability and Acrosome Integrity

Sperm viability was assessed by YO-PRO-1/IP (Figure 4). Results did not show significant changes in samples incubated with or without oxidant solution. Nevertheless, a slight decrease in viable cells was noted within the time, likely caused by damage on membrane spermatozoa. After a 4 h-incubation period, free VE treatment asserted a significant decrease on viable cells (Figure 4a), which were even more evident after 8 h of treatment. The same pattern within the following 24 h of incubation was observed (NE at 24 nM (NE24) ≈ NE at 12 nM (NE12) ≈ NE at 6 nM (NE6) > Control >> Free VE). Under oxidative stress conditions, the same pattern was maintained at any time of incubation. The population of dead cells was higher in the case of free VE treatment, which was almost dismissed after NE treatment when comparing to the Control (*p* > 0.05) (Figure 4b).

On the other hand, acrosomal status (Figure 5) was considerably affected by time with or without treatments. The percentage of spermatozoa with intact acrosomes dropped from 65.40 ± 5.38% at 0 h to 10.57 ± 2.75% at 24 h. Free VE treatments showed the worst results at any time, while NE did not show significant differences regarding Control (with and without oxidative stress), showing the following pattern (NE12 ≈ NE24 ≈ Control ≈ NE6 >> Free VE).

### 3.4. Vitamin E Nanoemulsions Effects on Mitochondrial Activity

The percentage of spermatozoa with high mitochondrial membrane potential (ΔΨ_m_) is slowly reduced over time (Figure 6). Incubation time did not change the pattern of distribution for Mitotracker Deep Red fluorescence within the viable subpopulations (NE12 ≈ NE24 ≈ Control ≈ NE6 <<Free VE), with or without oxidative stress. Free VE at 1 mM was the only treatment that caused a significant decrease in mitochondrial activity for the non and oxidized samples (*p* < 0.01).

### 3.5. Vitamin E Nanoemulsions Effects on Intracellular ROS Production and Lipid Peroxidation

Figure 7 displays the results for intracellular ROS production and lipid peroxidation after the different treatments. In general, ROS production mimicked lipid peroxidation levels. In the presence of oxidative stress, intracellular ROS (Figure 6a) suddenly increased from 0 to 24 h in Control treatments, while all the treatments significantly reduced CM-H_2_DCFDA fluorescence signal below control levels at any time. Between antioxidants treatments and when submitted to oxidative stress, NE showed a superior protective effect regarding free VE at any time.

Lipid peroxidation levels followed the same pattern (Figure 7b). In oxidized samples, all antioxidants reduced BODIPY C11 signal, remaining close to Control at 0 h without oxidative stress. Under oxidative stress conditions, lipid peroxidation levels increased from 0 to 24 h. However, antioxidants treatments significantly reduced BODIPY C11 fluorescence signal below control levels at any time.

### 3.6. Vitamin E Nanoemulsions Effects on DNA Damage

Figure 8 comprises SCSA^®®^ results after antioxidant treatments in the presence of oxidative and non-oxidative stress. In the case of % of DNA fragmentation index (%DFI) (Figure 8a), the analysis indicated that after 4 h and 8 h of treatment with a 24 nM concentration of NE (NE24), %DFI increased, mainly under oxidative stress, compared to Control. Moreover, data showed that High DNA Stainability (HDS) (Figure 7b) was negatively affected by NE treatments but without significant differences compared to Control. Free VE resulted to be the best option to counteract HDS. Nevertheless, although the statistical analysis showed significant differences, the damaged reported by analyzing %DFI and HDS for ram was still below the values reported as problematic for fertility.

## 4. Discussion

The use of drug delivery systems assisted by nanotechnology is a tool to be pursued in the field of assisted reproduction. At present, many nanodevices has been extensively tested for multiple species with satisfactory results. Zinc oxide nanoparticles [36], hydrogels (HVE) [29], cerium oxide nanoparticles [37,38], and chitosan–dextran sulphate nanoparticles [39] are some representative successful examples. In this sense and to the best our knowledge, herein we report for the first time data relying on the use of VE loaded nanoemulsions for the protection of ram spermatozoa against oxidative stress.

First, NE were assessed as a tool for ram spermatozoa preservation. For this purpose, the effects of different concentrations of NE (6, 12 and 24 mM of VE) were studied on ram spermatozoa stored for 24 h at 37 °C. Our results ascertained the use of NE to maintain the balance between the production and the scavenging of ROS and lipid peroxidation to prevent oxidative stress.

Second, evaluation of kinetic parameters on sperm samples were accomplished to report improvements in sperm motility after VE treatments. The relationship between kinematic parameters and fertility has been correlated in several species such as boar [40], bull [41], ram [42], or stallion [43]. One of the main factors related to the loss of sperm motility is the use of organic solvents, such as ethanol, needed to overcome the low water solubility of antioxidants, such as vitamin E. In this scenario, the detrimental effect of ethanol on sperm samples has been described in red deer (*Cervus elaphus hispanicus*) [14,15,17,18], red snappers (*Lutjanus argentimaculatus*) [44], male albino rats (*Sprague Dawley*) [45], and guineas pigs (*Cavia porcellus*) [46]. Previously, our research group reported that red deer spermatozoa incubated for 24 h at 37 °C with vitamin E nanoemulsions improved sperm motility with respect to free vitamin E, with or without exogenous oxidative stress [22,29], by passing the use of organic solvents. In this regard, data were related to the total and progressive motility support treatment of NE with respect to the use of free VE, with and without oxidative stress. This beneficial effect was mainly evident after 4 h of incubation time for all the kinetic parameters studied. At 24 h, the overall kinetic quality of ram spermatozoa dropped in all the treatments. These results are particularly remarkable due to the fact that not all the nanocarriers used on spermatozoa protection maintained sperm motility and quality. As a representative example, Soares et al. reported a non-beneficial effect on post-thawing boar sperm quality after treatment of VE-loaded polymeric nanocapsules [47]. Our results are consistent with our previous works related to red deer spermatozoa [22,29], where VE nanodevices were able to protect the sperm samples against oxidative stress and avoid the loss of spermatozoa motility for 24 h at 37 °C.

Third, the protective effect of vitamin E against oxidative stress damage on spermatozoa has been reported previously in several species such as red deer [14,17,21,22,29], bulls [48,49], boars [50,51], or humans [52]. Overproduction of ROS during sperm handling is one of the major factors responsible for weakening the spermatozoa quality. The results obtained in the present study showed that ROS production and lipid peroxidation levels significantly decrease in all antioxidant-treated groups (NE and free VE) with respect to Control after 4 h of incubation at 37 °C under oxidative stress. Within the antioxidant treatment groups, NE showed better results in comparison with free VE with and without oxidative stress. Again, these results in ram spermatozoa are in accordance with our previous studies in red deer [22,29]. However, some minor differences were observed. While spermatozoa free VE showed a significantly detrimental effect with and without oxidative stress with respect to NE in red deer, these significant differences were not observed in ram sperm likely due to the nature of samples (fresh sample for ovine and post-thawed samples for red deer).

Finally and in accordance with our previous studies on red deer spermatozoa [22,29], mitochondrial activity was positively correlated with sperm motility, which suggested the dependence of sperm motility on the functional integrity of mitochondria. These results are particularly interesting due to the fact that NE was allowed to preserve the highest percentages of spermatozoa with active mitochondria in comparison with free VE. In terms of viability, NE treatments showed their benefits over 4 h in comparison with free VE because no significant differences with respect to Control were observed, with or without oxidative stress at any time. A similar behavior was ascertained for acrosome integrity. In contrast, the incubation with NE24 was related to higher DNA fragmentation on samples with and without oxidative stress. These significant differences in biological terms are negligible because they indicate only 2–3% more fragmentation with respect to the rest of treatments, with all of them having a low and acceptable percentage of DNA fragmentation. However, this effect was not observed in red deer samples before, likely due to the lower concentration of NE evaluated [22,29].

## 5. Conclusions

In conclusion, our data disclose the use of NE as a feasible tool to preserve ram spermatozoa against oxidative stress by maintaining integral motility. Taking all our results as a whole, this study provides evidence for the application of NE in the field of assisted reproduction and may pave the way for the finding of more efficient nanodevices for the controlled release of antioxidants.

## Figures and Tables

**Figure 1 antioxidants-11-01988-f001:**
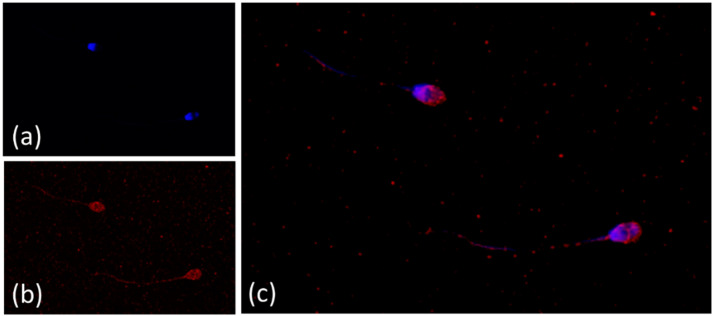
Vitamin E nanoemulsions on confocal microscopy. Ram spermatozoa stained with the fluorescent combination of Hoechst and DiD oil under a confocal laser scanning microscope. (**a**) Nucleus of sperm head stained by Hoechst 33342. (**b**) Vitamin E nanoemulsions labelled with DiD (DiIC_18_(5). (**c**) Spermatozoa marked with the fluorescent combination.

**Figure 2 antioxidants-11-01988-f002:**
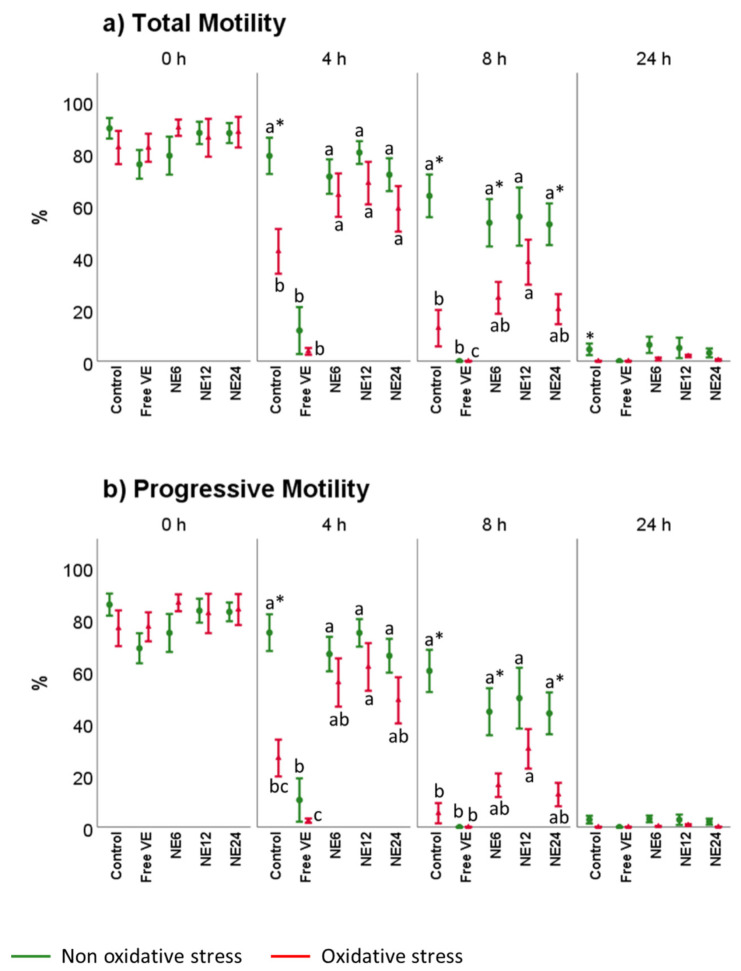
Vitamin E nanoemulsions effects on total and progressive sperm motility. Plots show the triple interaction of different treatments of vitamin E × oxidative treatment × incubation time for CASA-derived variables. The various lowercase Latin letters (*p* < 0.05) indicate significant differences between treatments of vitamin E, while asterisks (*p* < 0.05) compare each treatment at the same time with the different oxidative stress status.

**Figure 3 antioxidants-11-01988-f003:**
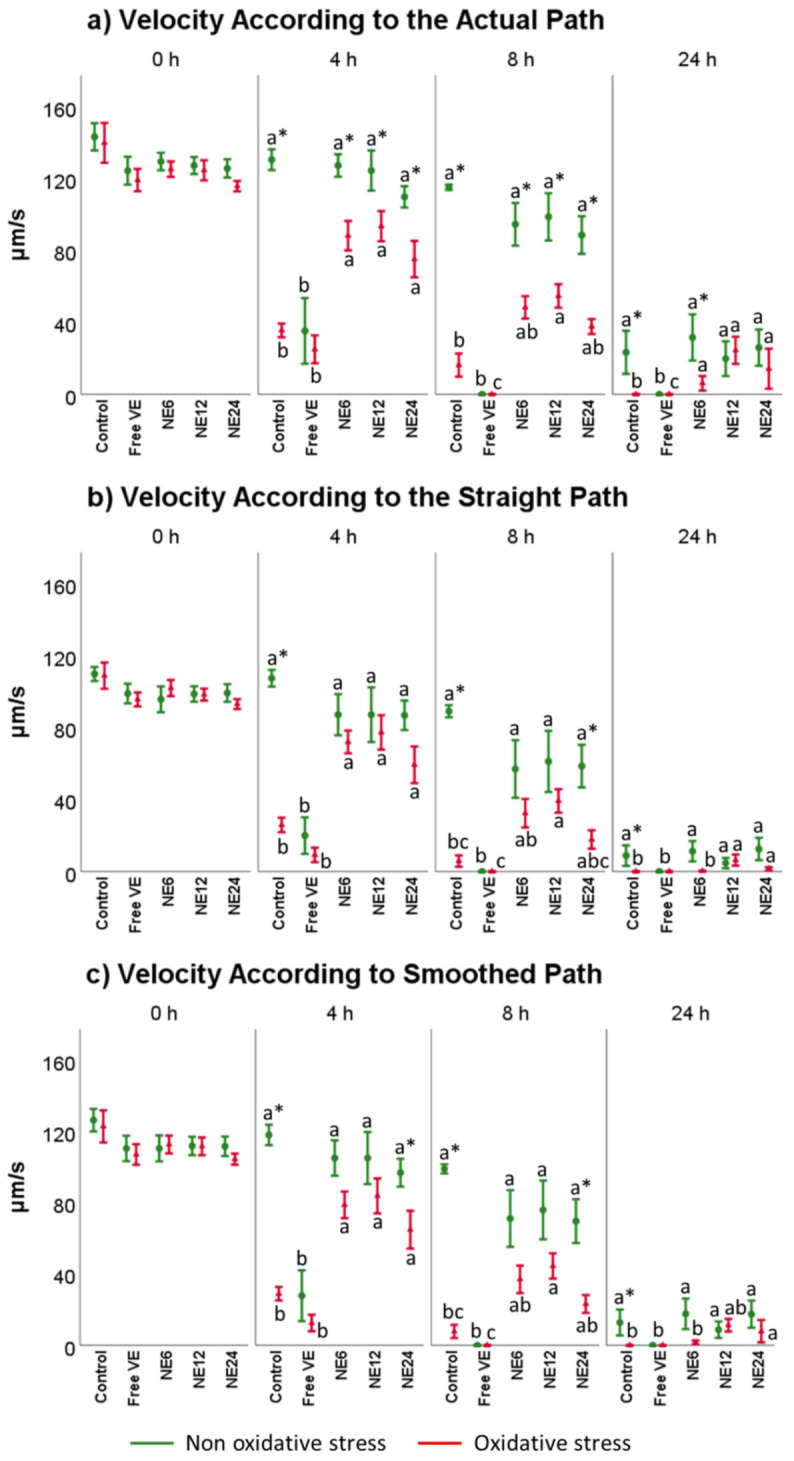
Vitamin E nanoemulsions effects on velocity sperm parameters. Plots show the triple interaction of different treatments of vitamin E × oxidative treatment × incubation time for CASA-derived variables. The various lowercase Latin letters (*p* < 0.05) indicate significant differences between treatments of vitamin E, while asterisks (*p* < 0.05) compare each treatment at the same time with the different oxidative stress status.

**Figure 4 antioxidants-11-01988-f004:**
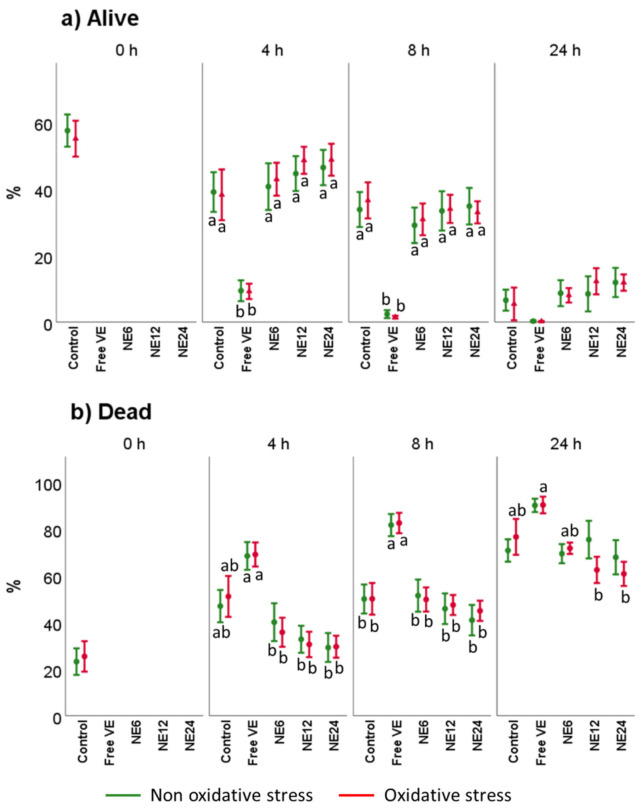
Vitamin E nanoemulsions effects on sperm viability. Plots show the triple interaction of different treatments of Vitamin E × oxidative treatment × incubation time for the flow cytometry analysis of sperm viability. The various lowercase Latin letters (*p* < 0.05) indicate significant differences between treatments of vitamin E.

**Figure 5 antioxidants-11-01988-f005:**
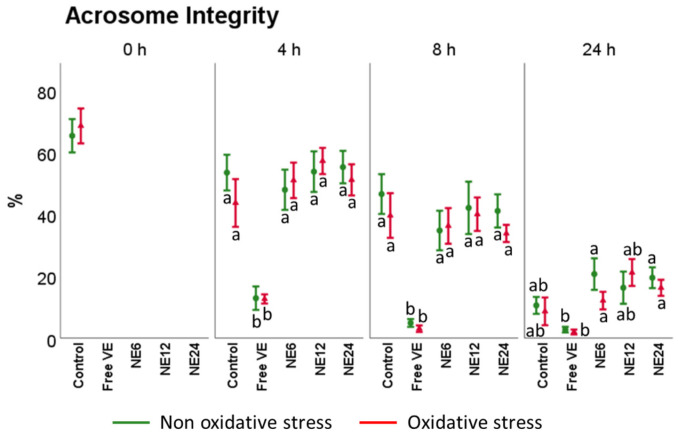
Vitamin E nanoemulsions effects on acrosome integrity. Plots show the triple interaction of different treatments of Vitamin E × oxidative treatment × incubation time for the flow cytometry analysis of mitochondrial activity (PNA−/PI−). The various lowercase Latin letters (*p* < 0.05) indicate significant differences between treatments of vitamin E, while different Greek letters (*p* < 0.05) compare each treatment at the same time with the different oxidative stress status.

**Figure 6 antioxidants-11-01988-f006:**
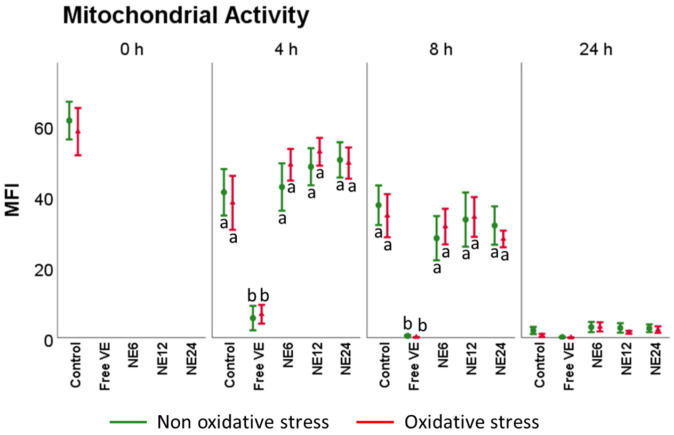
Vitamin E nanoemulsions effects on mitochondrial activity. Plots show the triple interaction of different treatments of Vitamin E × oxidative treatment × incubation time for the flow cytometry analysis of mitochondrial activity (YO-PRO-1−/Mitotracker deep red+). The various lowercase Latin letters (*p* < 0.05) indicate significant differences between treatments of vitamin E.

**Figure 7 antioxidants-11-01988-f007:**
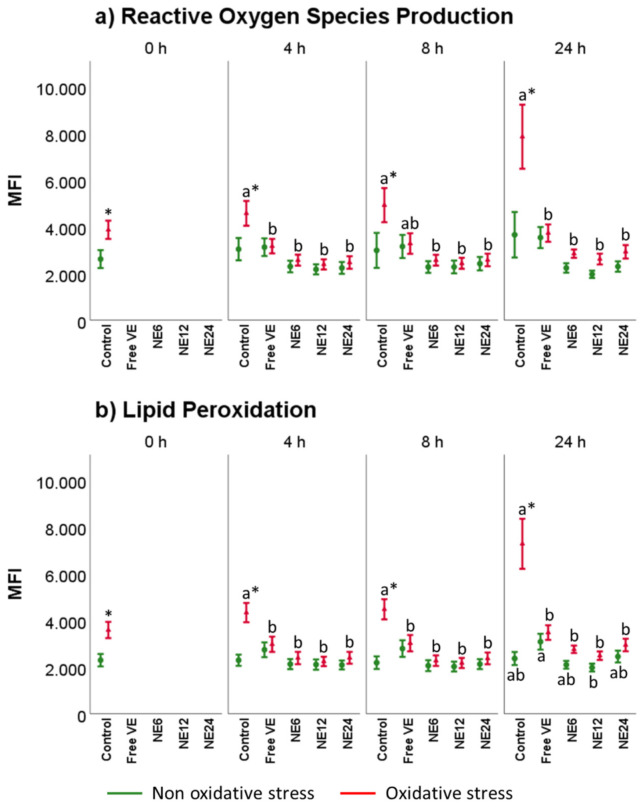
Vitamin E nanoemulsions effects on ROS production and lipid peroxidation. Plots show the triple interaction of different treatments of Vitamin E × oxidative treatment × incubation time for the flow cytometry analysis of reactive oxygen species (median fluorescence of H2DCFDA in PI− sperm) and lipid peroxidation (median green fluorescence of BODIPY C11). The various lowercase Latin letters (*p* < 0.05) indicate significant differences between treatments of vitamin E, while asterisks (*p* < 0.05) compare each treatment at the same time with the different oxidative stress status.

**Figure 8 antioxidants-11-01988-f008:**
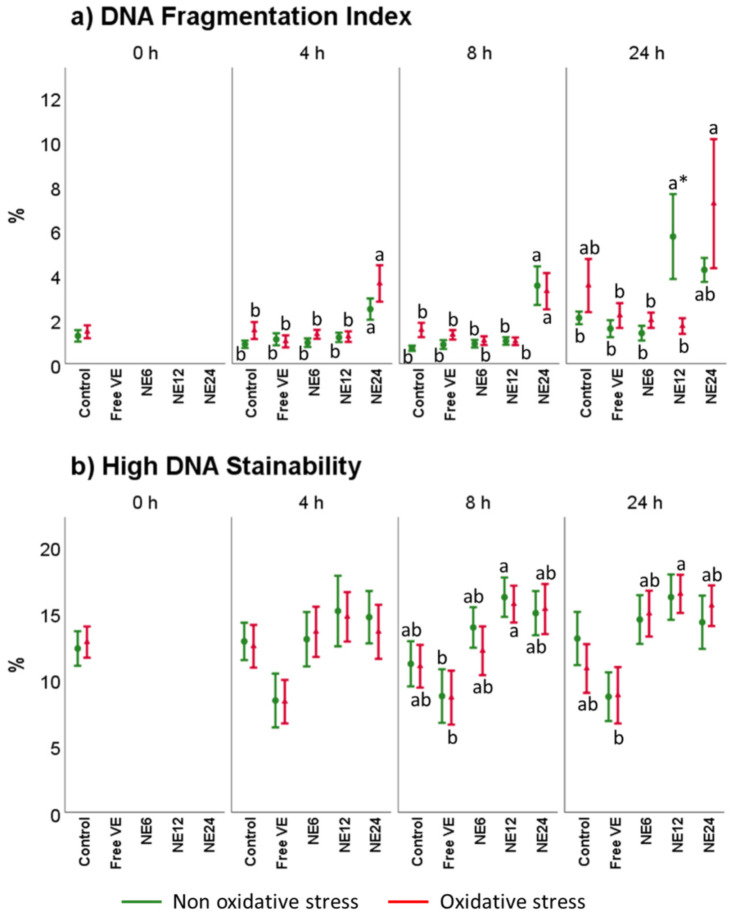
Vitamin E nanoemulsions effects on DNA. Plots represent the triple interaction of different treatments of Vitamin E × oxidative treatment × incubation time for the flow cytometry analysis of DNA Fragmentation Index and High DNA Stainability (median fluorescence of sperm assessed by SCSA^®®^). The various lowercase Latin letters (*p* < 0.05) indicate significant differences between treatments of vitamin E, while asterisks (*p* < 0.05) compare each treatment at the same time with the different oxidative stress status.

## Data Availability

The data presented in this study are available on request from the corresponding author. The data are not publicly available due to privacy.

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
