# Peer review of "Vitamin E Lipid-Based Nanodevices as a Tool for Ovine Sperm Protection against Oxidative Stress: Impact on Sperm Motility"

_antioxidants, 2022, doi:10.3390/antiox11101988_

Round 1
Reviewer 1 Report
In this manuscript the authors have evaluated the efficiency of vitamin E nanoemulsions (NE) on ram spermatozoa also submitted to oxidative stress. The manuscript is well written, the introduction is aimed at the topic, the materials and methods are exhaustive, the results well presented, even if the graphic representation can be improved, the discussion concise but clear. The topic is certainly of interest and the combination of spermatic activity and oxidative stress is a current research field. I suggest publication of the manuscript after the minor changes indicated.
Minor suggestions
Lines 11-15: these sentences are not necessary in the abstract.
Lines 125-127: please indicate the diluter of the trolox stock solution. In this section, two controls (white and VE) are indicated, while in the results the "white" is indicated as the control and the VE group as "free VE". I think we need to standardize.
Lines 165-167: please control the symbols +/- of acronyms.
Line 221: please indicate tests for normal distribution and homogeneity.
Figures 2-8: please avoid using the lines of conjunction between one group and another. This can lead to misunderstandings. Therefore, each group must be indicated with the symbol of the mean and the standard deviation.
Line 412-413: Can the authors hypothesize why the higher NE (NE24) concentration leads to high DNA damage?
Author Response
Response to reviewer 1
First, we want to thank all of you for your time and suggestions. We have tried to include all the corrections that you have highlighted in our previous file.
- Lines 11-15: these sentences are not necessary in the abstract.
- We have changed these sentences in the abstract.
- Lines 125-127: please indicate the diluter of the trolox stock solution. In this section, two controls (white and VE) are indicated, while in the results the "white" is indicated as the control and the VE group as "free VE". I think we need to standardize.
- The diluter for Trolox stock solution is indicated in the second paragraph of reagents and media (Material and methods, 2.1). In addition, we have standardized the labelling of different treatments.
- Lines 165-167: please control the symbols +/- of acronyms.
- We have controlled the symbols to avoid misunderstanding.
- Line 221: please indicate tests for normal distribution and homogeneity.
- We have indicated the different tests used.
- Figures 2-8: please avoid using the lines of conjunction between one group and another. This can lead to misunderstandings. Therefore, each group must be indicated with the symbol of the mean and the standard deviation.
- We have changed all the figures by eliminating the lines of conjunction between one group and another, as advised by the reviewer. However, we would prefer to keep the original graphs because, in our opinion, the results are more clearly presented. The lines of conjunction between groups help to display the results in a more visual and “reader-friendly” way. However, we leave it up to the Editor´s discretion.
- Line 412-413: Can the authors hypothesize why the higher NE (NE24) concentration leads to high DNA damage?
- We have hypothesized about this result in the test.
Response to reviewer 2
- The manuscript has an appropriate research design with materials and methods clearly described interesting results and is appropriate for publication as it is.
- Thank you so much for appreciation and your time.
Reviewer 2 Report
The manuscript entitled «Vitamin E Lipid-based nanodevices as a tool for ovine sperm protection against oxidative stress: Impact on sperm motility” by Jurado-Campos et al is a study on the efficiency of Vitamin E nanoemulsions (NE) (6, 12 , 24mM) on fresh ram spermatozoa under oxidative stress. The authors concluded that the use of these nanodevices could be a feasible tool to preserve ram spermatozoa against oxidative stress induced by the organic solvents required for lipid-soluble antioxidants by maintaining integral motility. Finally, the application of NE in the field of assisted reproduction is suggested.
The manuscript has an appropriate research design with materials and methods clearly described interesting results and is appropriate for publication as it is.
Author Response

(The authors gave the same response as above.)
